# Graph Cross Networks with Vertex Infomax Pooling

**Maosen Li**
Shanghai Jiao Tong University
maosen_li@sjtu.edu.cn

**Siheng Chen**[✉]
Shanghai Jiao Tong University
sihengc@sjtu.edu.cn

**Ya Zhang**[✉]
Shanghai Jiao Tong University
ya_zhang@sjtu.edu.cn

**Ivor Tsang**
Australian Artificial Intelligence Institute
University of Technology Sydney
Ivor.Tsang@uts.edu.au

## Abstract

We propose a novel *graph cross network* (GXN) to achieve comprehensive feature learning from multiple scales of a graph. Based on trainable hierarchical representations of a graph, GXN enables the interchange of intermediate features across scales to promote information flow. Two key ingredients of GXN include a novel *vertex infomax pooling* (VIPool), which creates multiscale graphs in a trainable manner, and a novel feature-crossing layer, enabling feature interchange across scales. The proposed VIPool selects the most informative subset of vertices based on the neural estimation of mutual information between vertex features and neighborhood features. The intuition behind is that a vertex is informative when it can maximally reflect its neighboring information. The proposed feature-crossing layer fuses intermediate features between two scales for mutual enhancement by improving information flow and enriching multiscale features at hidden layers. The cross shape of feature-crossing layer distinguishes GXN from many other multiscale architectures. Experimental results show that the proposed GXN improves the classification accuracy by $2.12\%$ and $1.15\%$ on average for graph classification and vertex classification, respectively. Based on the same network, the proposed VIPool consistently outperforms other graph-pooling methods.

## 1 Introduction

Recently, there are explosive interests in studying graph neural networks (GNNs) [32, 25, 50, 31, 12, 19, 55, 53, 9, 35, 10], which expand deep learning techniques to ubiquitous non-Euclidean graph data, such as social networks [52], bioinformatic networks [17], human activities [35] and motion interaction [28]. Achieving good performances on graph-related tasks, such as vertex classification [32, 25, 50] and graph classification [19, 55, 53], GNNs learn patterns from both graph structures and vertex information with feature extraction in spectral domain [5, 13, 32] or vertex domain [25, 40, 50, 34, 36, 12, 3]. Nevertheless, most GNN-based methods learn features of graphs with fixed scales, which might underestimate either local or global information. To address this issue, multiscale feature learning on graphs enables capturing more comprehensive graph features for downstream tasks [6, 23, 38].

Multiscale feature learning on graphs is a natural generalization from multiresolution analysis of images, whose related techniques, such as wavelets and pyramid representations, have been well studied in both theory and practice [26, 48, 44, 1, 56]. However, this generalization is technically nontrivial. While hierarchical representations and pixel-to-pixel associations across scales are

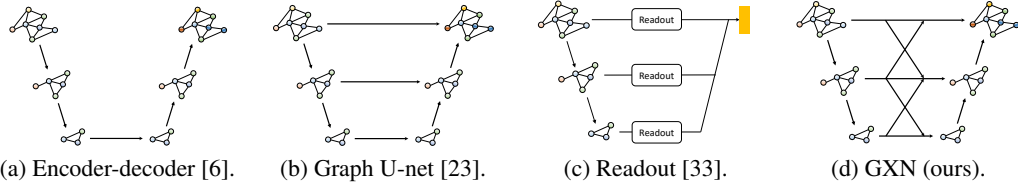

| (a) Encoder-decoder [6]. | (b) Graph U-net [23]. | (c) Readout [33]. | (d) GXN (ours). |

Figure 1: Architectures of multiscale graph neural networks. Our architecture adopts intermediate fusion.

straightforward for images with regular lattices, the highly irregular structures of graphs cause challenges in producing graphs at various scales [8] and aggregating features across scales.

To generate multiscale graphs, graph pooling methods are essential to compress large graphs into smaller ones. Conventional graph pooling methods [8, 45] leverage graph sampling theory and designed rules. Recently, some data-driven pooling methods are proposed, which automatically merge a fine-scale vertex subset to a coarsened vertex [15, 13, 46, 43, 37, 53]. The coarsened graph, however, might not have direct vertex-to-vertex association with the original scale. Some other graph pooling methods adaptively select vertices based on their importance over the entire graph [23, 33]; however, they fail to consider local information.

To aggregate features across multiple scales, existing attempts build encoder-decoder architecture [6, 37, 14] to learn graph features from the latent spaces, which might underestimate fine-scale information. Some other works gather the multiscale features in parallel and merge them as the final representation [38, 23, 22, 33], which might limit information flow across scales.

In this work, we design a new graph neural network to achieve multiscale feature learning on graphs, and our technical contributions are two-folds: a novel graph pooling operation to preserve informative vertices and a novel model architecture to exploit rich multiscale information.

**A novel graph pooling operation: Vertex infomax pooling (VIPool).** We propose a novel graph pooling operation by selecting and preserving those vertices that can maximally express their corresponding neighborhoods. The criterion of vertex-selection is based on the neural estimation of mutual information [2, 27, 51] between vertex and neighborhood features, thus we call the proposed pooling mechanism *vertex infomax pooling* (VIPool). Based on VIPool, we can implement graph pooling and unpooling to coarsen and refine multiple scales of a graph. Compared to the vertex-grouping-based methods [15, 13, 46, 43, 37, 53], the proposed VIPool provides the direct vertex-vertex association across scales and makes the coarsened graph structure and information fusion easier to achieve. Compared to other vertex-selection-based methods [23, 33], VIPool considers both local and global information on graphs by learning both vertex representation and graph structures.

**A novel model architecture: Graph cross network (GXN).** We propose a new model with a novel architecture called *graph cross network* (GXN) to achieve feature learning on multiscale graphs. Employing the trainable VIPool, our model creates multiscale graphs in data-driven manners. To learn features from all parallel scales, our model is built with a pyramid structure. To further promote information flow, we propose novel intermediate *feature-crossing layers* to interchange features across scales in each network layer. The intuition of feature-crossing is that the it improves information flow and exploits richer multiscale information in multiple network layers rather than only combine them in the last layer. Similar crossing structures have been explored for analyzing images [49, 48], but we cannot directly use those structures for irregular graphs. The proposed feature-crossing layer handles irregular graphs by providing the direct vertex-vertex associations across multiple graph scales and network layers; see typical multiscale architectures in Figure 1, where GXN is well distinguished because intermediate feature interchanging across scales forms a crossing shape.

*Remark*: In each individual scale, graph U-net [23] simply uses skip connections while GXN uses multiple graph propagation layers to extract features. The proposed feature-crossing layer is used to fuse intermediate features and cannot be directly applied to graph U-net.

To test our methods, we conduct extensive experiments on several standard datasets for both graph classification and vertex classification. Compared to state-of-the-art methods for these two tasks, GXN improves the average classification accuracies by 2.12% and 1.15%, respectively. Meanwhile, based on the same model architecture, our VIPool consistently outperforms previous graph pooling methods; and more intermediate connection leads to a better performance. [1]

## 2    Related Works

**Multiscale graph neural networks with graph pooling.** To comprehensively learn the multiscale graph representations, various multiscale network structures have been explored. Hierarchical encoder-decoder structures [6, 37, 14] learn graph features just from much coarse scales. LancozsNet [38] designs various graph filters on the multiscale graphs. Graph U-net [23] and readout functions [22, 33] design pyramid structures with skip-connections and combines features from all scales in the last layer. Compared to previous works, the proposed GXN has two main differences. 1) Besides the common late fusion of features, GXN uses intermediate fusion across scales, where the features at various scales in each network layer are fused to embed richer multiscale information. 2) GXN extracts hierarchical multiscale features through a deep network, previous Graph U-net [23] extracts features only once in each scale and then uses skip-connections to fuse feature across scales.

To compress a graph into multiple coarser scales, various methods of graph pooling are proposed. Early graph pooling methods are usually designed based on graph sampling theory [8] or graph coarsening [45]. With the study of deep learning, some works down-scale graphs in data-driven manner. The graph-coarsening-based pooling methods [15, 13, 46, 43, 37, 53, 54] cluster vertices and merge each cluster to a coarsened vertex; however, there is not vertex-to-vertex association to preserve the original vertex information. The vertex-selection-based pooling methods [23, 33] preserve selected vertices based on their importance, but tend to loss the original graph structures. Compared to previous works, the proposed VIPool in GXN is trained given an explicit optimization for vertex selection, and the pooled graph effectively abstracts the original graph structure.

**Mutual information estimation and maximization.** Given two variables, to estimate their mutual information whose extact value is hard to compute, some models are constructed based on the parameterization of neural networks. [2] leverages trainable networks to depict a lower bound of mutual information, which could be optimized toward a precise mutual information estimation. [27] maximizes the pixel-image mutual information to promote to capture the most informative image patterns via self-supervision. [51] maximizes the mutual information between a graph and each single vertex, where the representative vertex features are obtained. Similarly, [47] applies the mutual information maximization on graph classification. Compared to these mutual-information-based studies, the proposed VIPool, which also leverages mutual information maximization on graphs, aims to obtain an optimization for vertex selection by finding the vertices that maximally represent their local neighborhood. We also note that, in VIPool, the data distribution is defined on a single graph, while previous works [51, 47] assume to train on the distribution of multiple graphs.

## 3    Vertex Infomax Pooling

Before introducing the overall model, we first propose a new graph pooling method to create multiple scales of a graph. In this graph pooling, we select and preserve a ratio of vertices and connect them based on the original graph structure. Since downscaling graphs would lose information, it is critical to preserve as much information as possible in the pooled graph, which could maximally represent the original graphs. To this end, we propose a novel *vertex infomax pooling* (VIPool), preserving the vertices that carry high mutual information with their surrounding neighborhoods by mutual information estimation and maximization. The preserved vertices well represent local subgraphs, and they also abstract the overall graph structure based on a vertex selection criterion.

Mathematically, let $G(\mathcal{V}, \mathbf{A})$ be a graph with a set of vertices $\mathcal{V} = \{v_1, \dots, v_N\}$ whose features are $\mathbf{X} = [\mathbf{x}_1 \cdots \mathbf{x}_N]^\top \in \mathbb{R}^{N \times d}$, and an adjacency matrix $\mathbf{A} \in \{0, 1\}^{N \times N}$. We aim to select a subset $\Omega \subset \mathcal{V}$ that contains $|\Omega| = K$ vertices. Considering a criterion function $C(\cdot)$ to quantify the information of a vertex subset, we find the most informative subset through solving the problem,

$$\max_{\Omega \subset \mathcal{V}} \quad C(\Omega), \quad \text{subject to} \quad |\Omega| = K. \tag{1}$$

We design $C(\Omega)$ based on the mutual information between vertices and their corresponding neighborhoods, reflecting the vertices' abilities to express neighborhoods. In the following, we first introduce the computation of vertex-neighborhood mutual information, leading to the definition of $C(\cdot)$; we next select a vertex set by solving (1); we finally pool a fine graph based on the selected vertices.

**Mutual information neural estimation.** In a graph $G(\mathcal{V}, \mathbf{A})$, for any selected vertex $v$ in $\Omega \subset \mathcal{V}$, we define $v$'s neighborhood as $\mathcal{N}_v$, which is the subgraph containing the vertices in $\mathcal{V}$ whose geodesic distances to $v$ are no greater than a threshold $R$ according to the original $G(\mathcal{V}, \mathbf{A})$, i.e.

$\mathcal{N}_v = G(\{u\}_{d(u,v) \leq R}, \mathbf{A}_{\{u\},\{u\}})$. Let a random variable $\mathbf{v}$ be the feature of a randomly picked vertex in $\Omega$, the distribution of $\mathbf{v}$ is $P_{\mathbf{v}} = P(\mathbf{v} = \mathbf{x}_v)$, where $\mathbf{x}_v$ is the outcome feature value when we pick vertex $v$. Similarly, let a random variable $\mathbf{n}$ be the neighborhood feature associated with a randomly picked vertex in $\Omega$, the distribution of $\mathbf{n}$ is $P_{\mathbf{n}} = P(\mathbf{n} = \mathbf{y}_{\mathcal{N}_u})$, where $\mathbf{y}_{\mathcal{N}_u}$ is the outcome feature value when we pick vertex $u$'s neighborhood. The mutual information between selected vertices and neighborhoods is the KL-divergence between the joint distribution $P_{\mathbf{v},\mathbf{n}} = P(\mathbf{v} = \mathbf{x}_v, \mathbf{n} = \mathbf{y}_{\mathcal{N}_v})$ and the product of marginal distributions $P_{\mathbf{v}} \otimes P_{\mathbf{n}}$:

$$
\begin{aligned}
I^{(\Omega)}(\mathbf{v}, \mathbf{n}) &= D_{\mathrm{KL}}(P_{\mathbf{v},\mathbf{n}} || P_{\mathbf{v}} \otimes P_{\mathbf{n}}) \\
&\overset{(a)}{\geq} \sup_{T \in \mathcal{T}} \left\{ \mathbb{E}_{\mathbf{x}_v, \mathbf{y}_{\mathcal{N}_v} \sim P_{\mathbf{v},\mathbf{n}}} [T(\mathbf{x}_v, \mathbf{y}_{\mathcal{N}_v})] - \mathbb{E}_{\mathbf{x}_v \sim P_{\mathbf{v}}, \mathbf{y}_{\mathcal{N}_u} \sim P_{\mathbf{n}}} \left[ e^{T(\mathbf{x}_v, \mathbf{y}_{\mathcal{N}_u})-1} \right] \right\},
\end{aligned}
$$

where $(a)$ follows from $f$-divergence representation based on KL divergence [2]; $T \in \mathcal{T}$ is an arbitrary function that maps features of a pair of vertex and neighborhood to a real value, here reflecting the dependency of two features. To achieve more flexibility and convenience in optimization, $f$-divergence representation based on a non-KL divergence can be adopted [41], which still measures the vertex-neighborhood dependency. Here we consider a GAN-like divergence.

$$
I_{\mathrm{GAN}}^{(\Omega)}(\mathbf{v}, \mathbf{n}) \geq \sup_{T \in \mathcal{T}} \left\{ \mathbb{E}_{P_{\mathbf{v},\mathbf{n}}} [\log \sigma(T(\mathbf{x}_v, \mathbf{y}_{\mathcal{N}_v}))] + \mathbb{E}_{P_{\mathbf{v}}, P_{\mathbf{n}}} [\log(1 - \sigma(T(\mathbf{x}_v, \mathbf{y}_{\mathcal{N}_u})))] \right\},
$$

where $\sigma(\cdot)$ is the sigmoid function. In practice, we cannot go over the entire functional space $\mathcal{T}$ to evaluate the exact value of $I_{\mathrm{GAN}}^{(\Omega)}$. Instead, we parameterize $T(\cdot, \cdot)$ by a neural network $T_w(\cdot, \cdot)$, where the subscript $w$ denotes the trainable parameters. Through optimizing over $w$, we obtain a neural estimation of the GAN-based mutual information, denoted as $\widehat{I}_{\mathrm{GAN}}^{(\Omega)}$. We can define our vertex selection criterion function to be this neural estimation; that is,

$$
C(\Omega) = \widehat{I}_{\mathrm{GAN}}^{(\Omega)} = \max_w \frac{1}{|\Omega|} \sum_{v \in \Omega} \log \sigma(T_w(\mathbf{x}_v, \mathbf{y}_{\mathcal{N}_v})) + \frac{1}{|\Omega|^2} \sum_{(v,u) \in \Omega} \log(1 - \sigma(T_w(\mathbf{x}_v, \mathbf{y}_{\mathcal{N}_u}))).
$$

In $C(\Omega)$, the first term reflects the affinities between vertices and their own neighborhoods; and the second term reflects the differences between vertices and arbitrary neighborhoods. Notably, a higher $C$ score indicates that vertices maximally reflect their own neighborhoods and meanwhile minimally reflect arbitrary neighborhoods. To specify $T_w$, we consider $T_w(\mathbf{x}_v, \mathbf{y}_{\mathcal{N}_u}) = \mathcal{S}_w(\mathcal{E}_w(\mathbf{x}_v), \mathcal{P}_w(\mathbf{y}_{\mathcal{N}_u}))$, where the subscript $w$ indicates the associated functions are trainable[2], $\mathcal{E}_w(\cdot)$ and $\mathcal{P}_w(\cdot)$ are embedding functions of vertices and neighborhoods, respectively, and $\mathcal{S}_w(\cdot, \cdot)$ is an affinity function to quantify the affinity between vertices and neighborhoods; see an illustration in Figure 2. We implement $\mathcal{E}_w(\cdot)$ and $\mathcal{S}_w(\cdot, \cdot)$ by multi-layer perceptrons (MLPs), and implement $\mathcal{P}_w(\cdot)$ by aggregating vertex features and neighborhood connectivities in $\mathbf{y}_{\mathcal{N}_u}$; that is

$$
\mathcal{P}_w(\mathbf{y}_{\mathcal{N}_u}) = \frac{1}{R} \sum_{r=0}^{R} \sum_{\nu \in \mathcal{N}_u} \left( (\widetilde{\mathbf{D}}^{-1/2} \widetilde{\mathbf{A}} \widetilde{\mathbf{D}}^{-1/2})^r \right)_{\nu, u} \mathbf{W}^{(r)} \mathcal{E}_w(\mathbf{x}_\nu), \quad \forall u \in \Omega, \tag{2}
$$

where $\widetilde{\mathbf{A}} = \mathbf{A} + \mathbf{I} \in \{0,1\}^{N \times N}$ denotes the self-connected graph adjacency matrix and $\widetilde{\mathbf{D}}$ is the degree matrix of $\widetilde{\mathbf{A}}$; $\mathbf{W}^{(r)}$ is the trainable weight associated with the $r$th hop of neighbors; $\mathcal{P}_w(\cdot)$. The detailed derivation is presented in Appendix A.

When we maximize $C(\Omega)$ by training $\mathcal{E}_w(\cdot)$, $\mathcal{P}_w(\cdot)$ and $\mathcal{S}_w(\cdot, \cdot)$, we estimate the mutual information between vertices in $\Omega$ and their neighborhoods. This is similar to deep graph infomax (DGI) [51], which estimates the mutual information between any vertex feature and a global graph embedding. Both DGI and the proposed VIPool apply the techniques of mutual information neural estimation [2, 27] to the graph domain; however, there are two major differences. First, DGI aims to train a graph embedding function while VIPool aims to evaluate the importance of a vertex via its affinity to its neighborhood. Second, DGI considers the relationship between a vertex and an entire graph while VIPool learns the dependency between a vertex and a neighborhood. By varying the neighbor-hop $R$ of $\mathcal{N}_u$ in Eq. (2), VIPool is able to tradeoff local and global information.

**Solutions for vertex selection.** To solve the vertex selection problem (1), we consider the submodularity of mutual information [11] and employ a greedy algorithm: we select the first vertex with

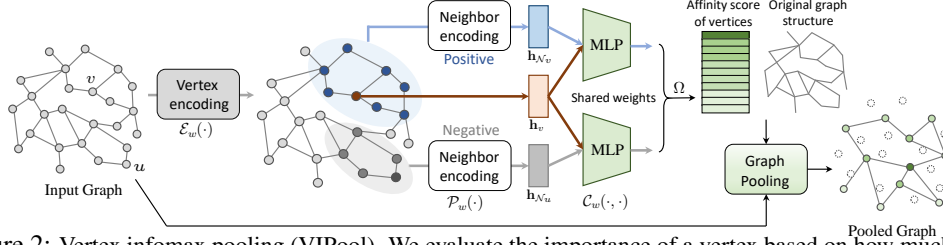

Figure 2: Vertex infomax pooling (VIPool). We evaluate the importance of a vertex based on how much it can reflect its own neighborhood and how much it can discriminate from an arbitrary neighborhood.

maximum $C(\Omega)$ with $|\Omega| = 1$; and we next add a new vertex sequentially by maximizing $C(\Omega)$ greedily; however, it is computationally expensive to evaluate $C(\Omega)$ for two reasons: (i) for any vertex set $\Omega$, we need to solve an individual optimization problem; and (ii) the second term of $C(\Omega)$ includes all the pairwise interactions involved with quadratic computational cost. To address issue (i), we set the vertex set to all the vertices in the graph, maximize $\widehat{I}_{\text{GAN}}^{(\mathcal{V})}$ to train $\mathcal{E}_w(\cdot)$, $\mathcal{P}_w(\cdot)$ and $\mathcal{S}_w(\cdot, \cdot)$. We then fix those three functions and evaluate $\widehat{I}_{\text{GAN}}^{(\Omega)}$. To address issue (ii), we perform negative sampling, approximating the second term [39], where we sample negative neighborhoods $\mathcal{N}_u$ from the entire graph, whose number equals the number of positive vertex samples; that is, $|\Omega|$.

**Graph pooling and unpooling.** After solving problem. (1), we obtain $\Omega$ that contains $K$ unique vertices selected from $\mathcal{V}$. To implement *graph pooling*, we further consider the distinct importance of different vertices in $\Omega$, we compute an affinity score for each vertex based on its ability to describe its neighborhood. For vertex $v$ with feature $\mathbf{x}_v$ and neighborhood feature $\mathbf{y}_{\mathcal{N}_v}$, the affinity score is

$$a_v = \sigma\left(\mathcal{S}_w(\mathcal{E}_w(\mathbf{x}_v), \mathcal{P}_w(\mathbf{y}_{\mathcal{N}_v}))\right) \in [0, 1], \quad \forall v \in \Omega. \tag{3}$$

Eq. (3) considers the affinity only between a vertex and its own neighborhood, showing the degree of vertex-neighborhood information dependency. We collect $a_v$ for $\forall v \in \Omega$ to form an affinity vector $\mathbf{a} \in [0, 1]^K$. For graph data pooling, the pooled vertex feature $\mathbf{X}_\Omega = \mathbf{X}(\text{id}, :) \odot (\mathbf{a}\mathbf{1}^\top) \in \mathbb{R}^{K \times d}$, where id denotes selected vertices's indices that are originally in $\mathcal{V}$, $\mathbf{1}$ is an all-one vector, and $\odot$ denotes the element-wise multiplication. With the affinity vector $\mathbf{a}$, we assign an importance to each vertex and provide a path for back-propagation to flow gradients. As for graph structure pooling, we calculate $\mathbf{A}_\Omega = \text{Pool}_A(\mathbf{A})$, and we consider three approaches to implement $\text{Pool}_A(\cdot)$:

• Edge removal, i.e. $\mathbf{A}_\Omega = \mathbf{A}(\text{id}, \text{id})$. This is simple, but loses significant structural information;

• Kron reduction [18], which is the Schur complement of the graph Laplacian matrix and preserves the graph spectral properties, but it is computationally expensive due to the matrix inversion;

• Cluster-connection, i.e. $\mathbf{A}_\Omega = \mathbf{S}\mathbf{A}\mathbf{S}^\top$ with $\mathbf{S} = \text{softmax}(\mathbf{A}(\text{id}, :)) \in [0, 1]^{K \times N}$. Each row of $\mathbf{S}$ represents the neighborhood of a selected vertex and the softmax function is applied for normalization. The intuition is to merge the neighboring information to the selected vertices [53].

Cluster-connection is our default implementation of the graph structure pooling. Figure 2 illustrates the overall process of vertex selection and graph pooling process.

To implement *graph unpooling*, inspired by [23], we design an inverse process against graph pooling. We initialize a zero matrix for the unpooled graph data, $\mathbf{X}' = \mathbf{O} \in \{0\}^{N \times d}$; and then, fill it by fetching the vertex features according to the original indices of retrained vertices; that is, $\mathbf{X}'(\text{id}, :) = \mathbf{X}_\Omega$. We then interpolate it through a graph propagation layer (implemented by graph convolution [32]) to propagate information from the vertices in $\Omega$ to the padded ones via the original graph structure.

## 4   Graph Cross Network

In this section, we propose the architecture of our *graph cross network* (GXN) for multiscale graph feature learning; see an exemplar model with 3 scales and 4 feature-crossing layers in Figure 3. The graph pooling/unpooling operations apply VIPool proposed in Section 3 and the graph propagation layers adopt the graph convolution layers [32]. A key ingredient of GXN is that we design *feature-crossing layers* to enhance multiscale information fusion. The entire GXN includes three stages: multiscale graphs generation, multiscale features extraction and multiscale readout.

**Multiscale graphs generation.** Given an input graph $G(\mathcal{V}, \mathbf{A})$ with vertex features, $\mathbf{X} \in \mathbb{R}^{N \times d}$, we aim to create graph representations at multiple scales. We first employ a graph propagation layer on

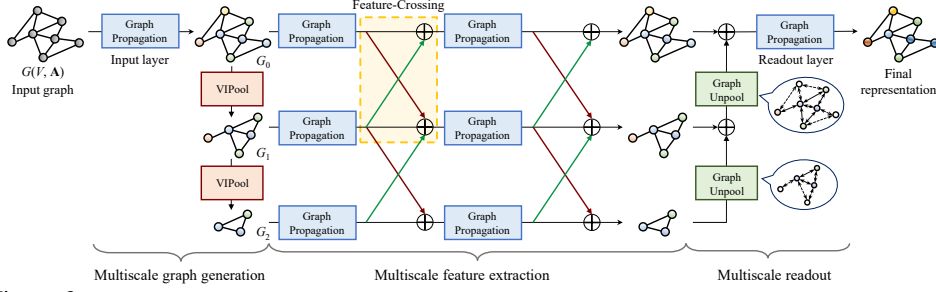

Figure 3: GXN architecture. We show an exemplar model with 3 scales and 4 feature-crossing layers.

the input graph to initially embed the finest scale of graph as $G_0(\mathcal{V}_0, \mathbf{A}_0)$ with $\mathcal{V}_0 = \mathcal{V}$, $\mathbf{A}_0 = \mathbf{A}$ and vertex representations $\mathbf{X}_0$, where the graph propagation layer is implemented by a graph convolution layer [32]. We then recursively apply VIPool for $S$ times to obtain a series of coarser scales of graph $G_1(\mathcal{V}_1, \mathbf{A}_1), \ldots, G_S(\mathcal{V}_S, \mathbf{A}_S)$ and corresponding vertex features $\mathbf{X}_1, \ldots, \mathbf{X}_S$ from $G_0$ and $\mathbf{X}_0$, respectively, where $|\mathcal{V}_s| > |\mathcal{V}_{s'}|$ for $\forall\, 1 \leq s < s' \leq S$.

**Multiscale features extraction.** Given multiscale graphs, we build a graph neural network at each scale to extract features. Each network consists of a sequence of graph propagation layers. To further enhance the information flow across scales, we propose feature-crossing layers between two consecutive scales at various network layers, which allows multiscale features to communicate and merge in the intermediate layers. Mathematically, at scale $s$ and the any network layer, let the feature matrix of graph $G_s$ be $\mathbf{X}_{\mathrm{h},s}$, the feature of graph $G_{s-1}$ after pooling be $\mathbf{X}_{\mathrm{h},s}^{(\mathrm{p})}$, and the feature of graph $G_{s+1}$ after unpooling be $\mathbf{X}_{\mathrm{h},s}^{(\mathrm{up})}$, the obtained vertex feature $\mathbf{X}'_{\mathrm{h},s}$ is formulated as

$$\mathbf{X}'_{\mathrm{h},s} = \mathbf{X}_{\mathrm{h},s} + \mathbf{X}_{\mathrm{h},s}^{(\mathrm{p})} + \mathbf{X}_{\mathrm{h},s}^{(\mathrm{up})}, \quad 0 < s < S.$$

For $s = 0$ or $S$, $\mathbf{X}_{\mathrm{h},s}$ is not fused by features from finer or coarser scales; see Figure 3. The graph pooling/unpooling here uses the same vertices as those obtained in multiscale graph generation to associate the vertices at different layers, but the affinity score $\mathbf{a}$ in each feature-crossing layer is trained independently to reflect the vertex importance at different levels. Note that the vertex-to-vertex association across scales is important here for feature-crossing and VIPool nicely fits it.

**Multiscale readout.** After multiscale feature extraction, we combine deep features at all the scales together to obtain the final representation. To align features at different scales, we adopt a sequence of graph unpooling operations implemented in the VIPool to transform all features to the original scale; see Figure 3. We finally leverage a readout graph propagation layer to further embed the fused multiscale features and generate the readout graph representation for various downstream tasks. In this work, we consider both graph classification and vertex classification.

**Model Training.** To train GXN, we consider the training loss with two terms: a graph-pooling loss $\mathcal{L}_{\mathrm{pool}} = -\widehat{I}_{\mathrm{GAN}}^{(\mathcal{V})}$ and a task-driven loss $\mathcal{L}_{\mathrm{task}}$. For graph classification, the task-driven loss is the cross-entropy loss between the predicted and ground-truth graph labels, $\mathcal{L}_{\mathrm{task}} = -\mathbf{y}^{\top} \log(\hat{\mathbf{y}})$, where $\mathbf{y}$ and $\hat{\mathbf{y}}$ are ground-truth label and predicted label of a graph; for vertex classification, it is the cross-entropy loss between the predictions and ground-truth vertex labels, $\mathcal{L}_{\mathrm{task}} = -\sum_{v \in \mathcal{V}_L} \mathbf{y}_v^{\top} \log(\hat{\mathbf{y}}_v)$, where $\mathbf{y}_v$ and $\hat{\mathbf{y}}_v$ are ground-truth and predicted vertex labels, and $\mathcal{V}_L$ contains labeled vertices. We finally define the overall loss as $\mathcal{L} = \mathcal{L}_{\mathrm{task}} + \alpha \mathcal{L}_{\mathrm{pool}}$, where the hyper-parameter $\alpha$ linearly decays per epoch from 2 to 0 during training, balancing a final task and vertex pooling[3].

## 5  Experimental Results

### 5.1  Datasets and Experiment Setup

**Datasets.** To test our GXN, we conduct extensive experiments for graph classification and vertex classification on several datasets. For **graph classification**, we use social network datasets: IMDB-B, IMDB-M and COLLAB [52], and bioinformatic datasets: D&D [17], PROTEINS [21], and ENZYMES [4]. Table 1 shows the dataset information. Note that no vertex feature is provided in three social network datasets, and we use one-hot vectors to encode the vertex degrees as vertex

Table 1: Graph classification accuracies (%) of different methods on different datasets. GXN (gPool) and GXN (SAGPool) denote that we apply previous pooling operations, gPool [23] and SAGPool [33] in our GXN framework, respectively. Various fashions of feature-crossing are presented, including fusion of coarse-to-fine ($\uparrow$), fine-to-coarse ($\downarrow$), no feature-crossing (noCross), and feature-crossing at early, late and all layers of networks.

| Dataset | IMDB-B | IMDB-M | COLLAB | D&D | PROTEINS | ENZYMES |
|---|---|---|---|---|---|---|
| # Graphs (Classes) | 1000 (2) | 1500 (3) | 5000 (3) | 1178 (2) | 1113 (2) | 600 (6) |
| Avg. # Vertices | 19.77 | 13.00 | 74.49 | 284.32 | 39.06 | 32.63 |
| PatchySAN [40] | $76.27 \pm 2.6$ | $69.70 \pm 2.2$ | $43.33 \pm 2.8$ | $72.60 \pm 2.2$ | $75.00 \pm 2.8$ | - |
| ECC [46] | $67.70 \pm 2.8$ | $43.48 \pm 3.0$ | $67.82 \pm 2.4$ | $72.57 \pm 4.1$ | $72.33 \pm 3.4$ | $29.50 \pm 7.6$ |
| Set2Set [24] | - | - | 71.75 | 78.12 | 74.29 | 60.15 |
| DGCNN [55] | $69.20 \pm 3.0$ | $45.63 \pm 3.4$ | $71.22 \pm 1.9$ | $76.59 \pm 4.1$ | $72.37 \pm 3.4$ | $38.83 \pm 5.7$ |
| DiffPool [53] | $68.40 \pm 6.1$ | $45.62 \pm 3.4$ | $74.83 \pm 2.0$ | $75.05 \pm 3.4$ | $73.72 \pm 3.5$ | $\mathbf{61.83 \pm 5.3}$ |
| Graph U-Net [23] | $73.40 \pm 3.7$ | $50.27 \pm 3.4$ | $77.58 \pm 1.6$ | $82.14 \pm 3.0$ | $77.20 \pm 4.3$ | $50.33 \pm 6.3$ |
| SAGPool [33] | $72.80 \pm 2.3$ | $49.43 \pm 2.6$ | $76.92 \pm 1.6$ | $78.35 \pm 3.5$ | $78.28 \pm 4.0$ | $52.67 \pm 5.8$ |
| AttPool [29] | $74.30 \pm 2.4$ | $50.67 \pm 2.7$ | $77.04 \pm 1.3$ | $79.20 \pm 3.8$ | $76.50 \pm 4.2$ | $55.33 \pm 6.2$ |
| GXN | $\mathbf{78.60 \pm 2.3}$ | $\mathbf{55.20 \pm 2.5}$ | $\mathbf{78.82 \pm 1.4}$ | $\mathbf{82.68 \pm 4.1}$ | $\mathbf{79.91 \pm 4.1}$ | $57.50 \pm 6.1$ |
| GXN (gPool) | $76.40 \pm 2.6$ | $53.62 \pm 2.6$ | $77.73 \pm 1.1$ | $81.94 \pm 4.3$ | $78.44 \pm 3.8$ | $55.43 \pm 5.3$ |
| GXN (SAGPool) | $76.90 \pm 2.4$ | $53.45 \pm 2.8$ | $78.10 \pm 1.5$ | $82.16 \pm 4.2$ | $78.58 \pm 4.1$ | $56.18 \pm 6.1$ |
| GXN (AttPool) | $77.30 \pm 2.3$ | $54.71 \pm 2.9$ | $78.22 \pm 1.2$ | $82.43 \pm 3.9$ | $78.09 \pm 4.3$ | $56.33 \pm 5.8$ |
| GXN ($\uparrow$) | $77.10 \pm 2.2$ | $54.22 \pm 2.5$ | $78.16 \pm 1.2$ | $82.03 \pm 3.8$ | $78.87 \pm 3.8$ | $56.83 \pm 5.8$ |
| GXN ($\downarrow$) | $76.60 \pm 2.1$ | $54.08 \pm 2.3$ | $77.68 \pm 1.0$ | $81.67 \pm 4.0$ | $78.64 \pm 4.2$ | $55.33 \pm 5.6$ |
| GXN (noCross) | $75.80 \pm 2.1$ | $52.68 \pm 2.7$ | $76.95 \pm 1.4$ | $81.23 \pm 3.9$ | $78.26 \pm 3.9$ | $55.17 \pm 5.7$ |
| GXN (early) | $77.50 \pm 2.4$ | $54.27 \pm 2.6$ | $78.18 \pm 1.1$ | $82.43 \pm 4.0$ | $79.20 \pm 4.0$ | $56.67 \pm 5.4$ |
| GXN (late) | $76.40 \pm 2.0$ | $53.83 \pm 2.4$ | $77.48 \pm 1.5$ | $82.16 \pm 3.6$ | $79.03 \pm 4.2$ | $56.00 \pm 5.9$ |
| GXN (all) | $\mathbf{78.60 \pm 2.3}$ | $\mathbf{55.20 \pm 2.5}$ | $\mathbf{78.82 \pm 1.4}$ | $\mathbf{82.68 \pm 4.1}$ | $\mathbf{79.91 \pm 4.1}$ | $57.50 \pm 6.1$ |

features, explicitly utilizing some structural information. We use the same dataset separation as in [23], perform 10-fold cross-validation, and show the average accuracy for evaluation. For **vertex classification**, we use three classical citation networks: Cora, Citeseer and Pubmed [32]. We perform both full-supervised and semi-supervised vertex classification; that is, for full-supervised classification, we label all the vertices in training sets for model training, while for semi-supervised, we only label a few vertices (around 7% on average) in training sets. We use the default separations of training/validation/test subsets. See more information of all used datasets in Appendix.

**Model configuration.** We implement GXN with PyTorch 1.1 on one GTX-1080Ti GPU. For **graph classification**, we consider three scales, which preserve 50% to 100% vertices from the original scales, respectively. For both input and readout layers, we use 1-layer GCNs; for multiscale feature extraction, we use two GCN layers followed by ReLUs at each scale and feature-crossing layers between any two consecutive scales at any layers. After the readout layers, we unify the embeddings of various graphs to the same dimension by using the same SortPool in DGCNN [55], AttPool [29] and Graph U-Net [23]. In the VIPool, we use a 2-layer MLP and $R$-layer GCN ($R = 1$ or $2$) as $\mathcal{E}_w(\cdot)$ and $\mathcal{P}_w(\cdot)$, and use a linear layer as $\mathcal{S}_w(\cdot, \cdot)$. The hidden dimensions are 48. To improve the efficiency of solving problem (1), we modify $C(\Omega)$ by preserving only the first term. In this way, we effectively reduce the computational costs to sovle (1) from $\mathcal{O}(|V|^2)$ to $\mathcal{O}(|V|)$, and each vertex contributes the vertex set independently. The optimal solution is **top-$K$** vertices. We compare the outcomes of $C(\Omega)$ by the greedy algorithm and top-k method in Figure 5. For **vertex classification**, we use similar architecture as in graph classification, while the hidden feature are 128-dimension. We directly use the readout layer for vertex classification. In the loss function $\mathcal{L}$, $\alpha$ decays from 2 to 0 during training, where the VIPool needs fast convergence for vertex selection; and the model gradually focuses more on tasks based on the effective VIPool. We use Adam optimizer [16] and the learining rates range from 0.0001 to 0.001 for different datasets.

### 5.2 Comparison

**Graph classification.** We compare the proposed GXN to representative GNN-based methods, including PatchySAN [40], ECC [46], Set2Set [24], DGCNN [55], DiffPool [53], Graph U-Net [23], SAGPool [33], AttPool [29], and StructPool [54], where most of them performed multiscale graph feature learning. We unify the train/test data splits and processes of model selection for fair comparison [20]. Additionally, we design several variants of GXN: 1) to test the superiority of VIPool, we apply gPool [23], SAGPool [33] and AttPool [29] in the same architecture of GXN, denoted as 'GXN (gPool)', 'GXN (SAGPool)' and 'GXN (AttPool)', respectively; 2) we investigate different feature-crossing mechanism, including various crossing directions and crossing positions. Table 1 compares the accuracies of various methods for graph classification. We see that our model outperforms the state-of-the-art methods on 5 out of 6 datasets, achieving an improvement by 2.12% on

Table 2: Vertex classification accuracies (%) of different methods, where 'full-sup.' and 'semi-sup.' denote the scenarios of full-supervised and semi-supervised vertex classification, respectively.

| Dataset | Cora | | Citeseer | | Pubmed | |
|---|---|---|---|---|---|---|
| # Vertices (Classes) | 2708 (7) | | 3327 (6) | | 19717 (3) | |
| Supervision | full-sup. | semi-sup. | full-sup. | semi-sup. | full-sup. | semi-sup. |
| DeepWalk [42] | $78.4 \pm 1.7$ | $67.2 \pm 2.0$ | $68.5 \pm 1.8$ | $43.2 \pm 1.6$ | $79.8 \pm 1.1$ | $65.3 \pm 1.1$ |
| ChebNet [13] | $86.4 \pm 0.5$ | $81.2 \pm 0.5$ | $78.9 \pm 0.4$ | $69.8 \pm 0.5$ | $88.7 \pm 0.3$ | $74.4 \pm 0.4$ |
| GCN [32] | $86.6 \pm 0.4$ | $81.5 \pm 0.5$ | $79.3 \pm 0.5$ | $70.3 \pm 0.5$ | $90.2 \pm 0.3$ | $79.0 \pm 0.3$ |
| GAT [50] | $87.8 \pm 0.7$ | $83.0 \pm 0.7$ | $80.2 \pm 0.6$ | $73.5 \pm 0.7$ | $90.6 \pm 0.4$ | $79.0 \pm 0.3$ |
| FastGCN [7] | $85.0 \pm 0.8$ | $80.8 \pm 1.0$ | $77.6 \pm 0.8$ | $69.4 \pm 0.8$ | $88.0 \pm 0.6$ | $78.5 \pm 0.7$ |
| ASGCN [30] | $87.4 \pm 0.3$ | - | $79.6 \pm 0.2$ | - | $90.6 \pm 0.3$ | - |
| Graph U-Net [23] | - | 84.4 | - | 73.2 | - | 79.6 |
| GXN | $\mathbf{88.9 \pm 0.4}$ | $\mathbf{85.1 \pm 0.6}$ | $\mathbf{80.9 \pm 0.4}$ | $\mathbf{74.8 \pm 0.4}$ | $\mathbf{91.8 \pm 0.3}$ | $\mathbf{80.2 \pm 0.3}$ |
| GXN (noCross) | $87.3 \pm 0.4$ | $83.2 \pm 0.5$ | $79.5 \pm 0.4$ | $73.7 \pm 0.3$ | $91.1 \pm 0.2$ | $79.6 \pm 0.3$ |

average accuracies. Besides, VIPool and more feature-crossing lead to better performance. We also show the qualitative results of vertex selection of different graph pooling methods in Appendix.

**Vertex classification.** We compare GXN to state-of-the-art methods: DeepWalk [42], GCN [32], GraphSAGE [25], FastGCN [7], ASGCN [30], and Graph U-Net [23] for vertex classification. We reproduce these methods for both full-supervised and semi-supervised learning based on their official codes. Table 2 compares the vertex classification accuracies of various methods. Considering both full-supervised and semi-supervised settings, we see that our model achieves higher average accuracy by $1.15\%$. We also test a degraded GXN without any feature-crossing layer, and we see that the feature-crossing layers improves the accuracies by $1.10\%$ on average; see more results in Appendix.

## 5.3 Model Analysis

We further conduct detailed analysis about the GXN architecture and VIPool.

**GXN architectures.** We test the architecture with various graph scales and feature-crossing layers. Base on the dataset of IMDB-B for graph classification, we vary the number of graph scales from 1 to 5 and the number of feature-crossing layers from 1 to 3. We present the vertex classification results in Table 3. We see that the architecture with 3 graph scales and 2 feature-crossing layers leads to the best performance. Compared to use only one graph scale, using three graph scales significantly improve the graph classification accuracy by $4.38\%$ on average, indicating the importance of multiscale representations. When we use more than three scales, the classification results tend to be stable, indicating the redundant scales. To keep the model efficiency and effectiveness, we adopt three scales of graphs. As for the number of feature-crossing layers, only using one feature-crossing layer do not provide sufficient information for graph classification; while using more than two feature-crossing layers tends to damage model performance due to the higher model complexity.

**Hops of neighborhood in VIPool.** To validate the effects of different ranges of neighborhood information in VIPool, we vary the neighbor-hops $R$ in $\mathcal{P}_w(\cdot)$ from 1 to 5 and perform graph classification on D&D and IMDB-B. When $R$ increases, we push a vertex to represent a bigger neighborhood with more global information. Figure 4 shows the graph classification accuracies with various $R$ on the two datasets. We see that, for D&D, which include graphs with relatively larger sizes (see Table 1, line 3), when $R = 2$, the model achieves the best performance, reflecting that vertices could express their neighborhoods within $R = 2$; while for IMDB-B with smaller graphs, vertices tend to express their 1-hop neighbors better. This reflects that VIPool achieves a flexible trade-off between local and global information through varying $R$ to adapt to various graphs.

**Approximation of C function.** In VIPool, to optimize problem (1) more efficiently, we substitute the original $C(\Omega)$ by only preserving the positive term $C_+(\Omega) = \sum_{v \in \Omega} \log \sigma(\mathcal{S}_w(\mathbf{h}_v, \mathbf{h}_{\mathcal{N}_v}))$ and

| | | accuracy | | |
|---|---|---|---|---|
| | # cross | 1 | 2 | 3 |
| **# scales** | 1 | 75.30 | - | - |
| | 2 | 76.80 | 77.40 | 76.20 |
| | 3 | 77.50 | **78.60** | 77.70 |
| | 4 | 77.60 | 78.20 | 78.10 |
| | 5 | 77.50 | **78.60** | 77.90 |

Table 3: Graph classification accuracies (%) with various scales and feature-crossing layers on IMDB-B.

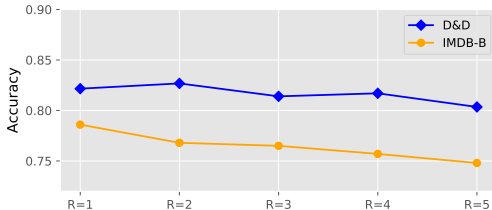

Figure 4: Graph classification accuracies (%) with neighbor-hops $R$ from 1 to 5 on D&D and IMDB-B.

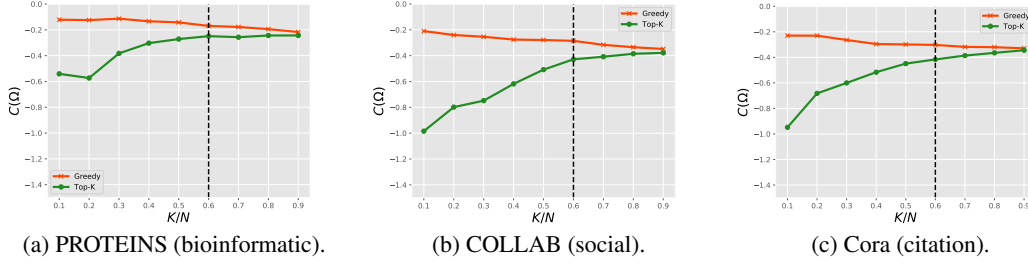

| (a) PROTEINS (bioinformatic). | (b) COLLAB (social). | (c) Cora (citation). |

Figure 5: Comparison of C values on different types of graph datasets.

Table 4: Vertex classification accuracies and time costs per epoch with different $\mathrm{Pool}_A(\cdot)$ on Cora.

| $\mathrm{Pool}_A(\cdot)$ | Accuracy (%) | Time (s) |
|---|---|---|
| Edge-Remove | 84.6 | 0.44 |
| Kron-Reduce | 85.3 | 8.36 |
| Clus-Connect | 85.1 | 1.28 |

Table 5: Graph classification accuracies and time costs per epoch with different $\mathrm{Pool}_A(\cdot)$ on IMDB-B.

| $\mathrm{Pool}_A(\cdot)$ | Accuracy (%) | Time (s) |
|---|---|---|
| Edge-Remove | 78.20 | 1.97 |
| Kron-Reduce | 78.50 | 13.44 |
| Clus-Connect | 78.60 | 3.75 |

maximize $C_+(\Omega)$ by selecting 'Top-K', which also obtains the optimal solution. To see the performance gap between the original and the accelerated versions, we compare the exact value of $C(\Omega)$ with the selected $\Omega$ by optimizing $C(\Omega)$ with greedy algorithm and optimizing $C_+(\Omega)$ with 'Top-K' method, respectively, on different types of datasets: bioinformatic, social and citation networks. We vary the ratio of the vertex selection among the global graph from 0.1 to 0.9. Figure 5 compares the $C(\Omega)$ as a function of selection ratio with two algorithms on 3 datasets, and the vertical dash lines denotes the boundaries where the value gaps equal to 10%. We see that, when we select small percentages (e.g. $< 60\%$) of vertices, the $C$ value obtained by the greedy algorithm is much higher than 'Top-K' method; when we select more vertices, there are very small gaps between the two optimization algorithms, indicating two similar solutions of vertex selection. In GXN, we set the selection ratio above 60% in each graph pooling. More results about the model performances varying with ratios of vertices selection are presented in Appendix.

**Graph structure pooling.** In VIPool, we consider three implementations for graph structure pooling $\mathrm{Pool}_A(\cdot)$: edge-removal, Kron reduction and cluster-connection. We test these three operations for semi-supervised vertex classification on Cora and graph classification on IMDB-B, and we show the classification accuracies and time costs of the three graph structure pooling operations (denoted as 'Edge-Remove', 'Kron-Reduce' and 'Clus-Connect', respectively) in Tables 4 and 5. We see that Kron reduction or cluster-connection tend to provide the best accuracies on different datasets, but Kron reduction is significantly more expensive than the other two methods due to the matrix inversion. On the other hand, cluster-connection provides a better tradeoff between effectiveness and efficiency and we thus consider cluster-connection as our default choice.

# 6 Conclusions

This paper proposes a novel model *graph cross network* (GXN), where we construct parallel networks for feature learning at multiple scales of a graph and design novel feature-crossing layers to fuse intermediate features across multiple scales. We propose vertex infomax pooling (VIPool), selecting those vertices that maximally describe their neighborhood information. Based on the selected vertices, we coarsen graph structures and the corresponding graph data. VIPool is optimized based on the neural estimation of the mutual information between vertices and neighborhoods. Extensive experiments show that (i) GXN outperforms most state-of-the-art methods on graph classification and vertex classification; (ii) VIPool outperforms the other pooling methods; and (iii) more intermediate fusion across scales leads to better performances. In the future, the proposed GXN architectures could be effectively applied on many multi-view or multi-modal graph learning tasks, which perform feature interaction for informative pattern capturing.

# Broader Impact of Our Work

In this work, we aim to propose a method for multiscale feature learning on graphs, achieving two basic but challenging tasks: graph classification and vertex classification. This work has the following potential impacts to the society and the research community.

This work could be effectively used in many practical and important scenarios such as drug molecular analysis, social network mining, biometrics, human action recognition and motion prediction, etc., making our daily life more convenient and efficient. Due to the ubiquitous graph data, in most cases, we can try to construct multiscale graphs to comprehensively obtain rich detailed, abstract, and even global feature representations, and effectively improve downstream tasks.

Our network structure can not only solve problem of feature learning with multiple graph scales, but also can be applied to the pattern learning of heterogeneous graphs, or other cross-modal or cross-view machine learning scenarios. This is of great significance for improving the ability of pattern recognition, feature transfer, and knowledge distillation to improve the computational efficiency.

At the same time, this work may have some negative consequences. For example, in social networks, it is uncomfortable even dangerous to use the models based on this work to over-mine the behavior of users, because the user's personal privacy and information security are crucial; companies should avoid mining too much users' personal information when building social platforms, keeping a safe internet environment.

## Acknowledgement and Disclosure of Funding

This work is supported by the National Key Research and Development Program of China (No. 2019YFB1804304), SHEITC (No. 2018- RGZN-02046), 111 plan (No. BP0719010), and STCSM (No. 18DZ2270700), and State Key Laboratory of UHD Video and Audio Production and Presentation. Prof. Ivor Tsang is supported by ARC DP180100106 and DP200101328.

## Footnotes

* This work was done while Siheng Chen was working at Mitsubishi Electric Research Laboratories (MERL).

[1] The code could be downloaded at `https://github.com/limaosen0/GXN`

[2] The trainable parameters in $\mathcal{E}_w(\cdot)$, $\mathcal{P}_w(\cdot)$, and $\mathcal{S}_w(\cdot, \cdot)$ are not weight-shared.

[3] VIPool is trained through both $\mathcal{L}_{\mathrm{pool}}$ and $\mathcal{L}_{\mathrm{task}}$, which makes graph pooling adapt to a specific task.

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
