[Supplementary Material]

# Graph Cross Networks with Vertex Infomax Pooling

**Maosen Li**
Shanghai Jiao Tong University
maosen_li@sjtu.edu.cn

**Siheng Chen**
Mitsubishi Electric Laboratories
schen@merl.com

**Ya Zhang**
Shanghai Jiao Tong University
ya_zhang@sjtu.edu.cn

**Ivor Tsang**
University of Technology, Sydney
Ivor.Tsang@uts.edu.au

## Appendix A. Mutual Information Neural Estimation for Vertex Selection

An individual vertex is fully identified through its feature, which works as the vertex attribute. Given an vertex set $\mathcal{V}$ that contains all the vertices on the graph and a vertex subset $\Omega \subset \mathcal{V}$ which contains the selected vertices, we let a random variable $\mathbf{v}$ to represent the vertex feature when we randomly pick a vertex from $\Omega$. Then we define the probability distribution of $\mathbf{v}$ as

$$P_{\mathbf{v}} = P(\mathbf{v} = \mathbf{x}_v), \quad \forall v \in \Omega$$

where $\mathbf{x}_v$ is the feature value when we pick vertex $v$.

The neighborhood of any vertex $u \in \Omega$ is defined as $\mathcal{N}_u$, which is the subgraph containing the vertices in $\mathcal{V}$ whose geodesic distances to the central vertex $u$ are no greater than a threshold $R$, i.e. $\mathcal{N}_v = G(\{u\}_{d(u,v) \le R}, \mathbf{A}_{\{u\},\{u\}})$. Let a random variable $\mathbf{n}$ be the neighborhood features when we randomly pick a central vertex from $\Omega$, then we define the probability distribution of $\mathbf{n}$ as

$$P_{\mathbf{n}} = P(\mathbf{n} = \mathbf{y}_{\mathcal{N}_u}), \quad \forall u \in \Omega$$

where $\mathbf{y}_{\mathcal{N}_u} = [\mathbf{A}_{\mathcal{N}_u,\mathcal{N}_u}, \{\mathbf{x}_\nu\}_{\nu \in \mathcal{N}_u}]$ denotes the neighborhood feature value when we pick vertex $u$'s neighborhood, including both the internal connectivity information and contained vertex features in the neighborhood $\mathcal{N}_u$.

Therefore, we define the joint distribution of the random variables of vertex features and neighborhood features, which is formulated as

$$P_{\mathbf{v},\mathbf{n}} = P(\mathbf{v} = \mathbf{x}_v, \mathbf{n} = \mathbf{y}_{\mathcal{N}_v}), \quad \forall v \in \Omega$$

where the joint distribution reflects probability that we randomly pick the corresponding vertex feature and neighborhood feature of the same vertex $v$ together.

The *mutual information* between the vertex features and the neighborhood features is defined as the KL-divergence between the joint distribution $P_{\mathbf{v},\mathbf{n}}$ and the product of the marginal distributions of two random variable, $P_{\mathbf{v}} \otimes P_{\mathbf{n}}$; that is

$$I^{(\Omega)}(\mathbf{v}, \mathbf{n}) = D_{\mathrm{KL}}\left(P_{\mathbf{v},\mathbf{n}} \| P_{\mathbf{v}} \otimes P_{\mathbf{n}}\right),$$

This mutual information measures of the mutual dependency between vertices and neighborhoods in the selected vertex subset $\Omega$. The KL divergence admits the $f$-representation [1],

$$D_{\mathrm{KL}}\left(P_{\mathbf{v},\mathbf{n}} \| P_{\mathbf{v}} \otimes P_{\mathbf{n}}\right) \ge \sup_{T \in \mathcal{T}} \left\{ \mathbb{E}_{\mathbf{x}_v,\mathbf{y}_{\mathcal{N}_v} \sim P_{\mathbf{v},\mathbf{n}}}[T(\mathbf{x}_v, \mathbf{y}_{\mathcal{N}_v})] - \mathbb{E}_{\mathbf{x}_v \sim P_{\mathbf{v}}, \mathbf{y}_{\mathcal{N}_u} \sim P_{\mathbf{n}}}\left[e^{T(\mathbf{x}_v, \mathbf{y}_{\mathcal{N}_u})-1}\right] \right\}, \quad (1)$$

where $\mathcal{T}$ is an arbitrary class of functions that maps a pair of vertex features and neighborhood features to a real value, and here we use $T(\cdot, \cdot)$ to compute the dependency of two features. It could be a tight lower-bound of mutual information if we search any possible function $T \in \mathcal{T}$.

Note that the main target here is to propose a vertex-selection criterion based on quantifying the dependency between vertices and neighborhood. Therefore instead of computing the exact mutual information based on KL divergence, we can use non-KL divergences to achieve favourable flexibility and convenience in optimization. Both non-KL and KL divergences can be formulated based on the same $f$-representation framework. Here we start from the general $f$-divergence between the joint distribution and the product of marginal distributions of vertices and neighborhoods.

$$D_f\left(P_{\mathbf{v},\mathbf{n}}||P_{\mathbf{v}} \otimes P_{\mathbf{n}}\right) \;=\; \int P_{\mathbf{v}} P_{\mathbf{n}} f\left(\frac{P_{\mathbf{v},\mathbf{n}}}{P_{\mathbf{v}} P_{\mathbf{n}}}\right) d\mathbf{x}_v d\mathbf{y}_{\mathcal{N}_v}$$

where $f(\cdot)$ is a convex and lower-semicontinuous divergence function. when $f(x) = x \log x$, the $f$-divergence is specified as KL divergence. The function $f(\cdot)$ has a convex conjugate function $f^*(\cdot)$, i.e. $f^*(t) = \sup_{x \in \mathrm{dom}_f}\{xt - f(x)\}$, where $\mathrm{dom}_f$ is the definition domain of $f(\cdot)$. Note that the two functions $f(\cdot)$ and $f^*(\cdot)$ is dual to each other. According to the Fenchel conjugate [6], the $f$-divergence can be modified as

$$\begin{aligned}
D_f(P_{\mathbf{v},\mathbf{n}}||P_{\mathbf{v}} \otimes P_{\mathbf{n}}) &= \int P_{\mathbf{x}} P_{\mathbf{n}} \sup_{t \in \mathrm{dom}_{f^*}} \left\{ t\frac{P_{\mathbf{x},\mathbf{n}}}{P_{\mathbf{v}} P_{\mathbf{n}}} - f^*(t) \right\} \\
&\geq \sup_{T \in \mathcal{T}} \left\{ \mathbb{E}_{P_{\mathbf{v},\mathbf{n}}}[T(\mathbf{x}_v, \mathbf{y}_{\mathcal{N}_v})] - \mathbb{E}_{P_{\mathbf{v}}, P_{\mathbf{n}}}[f^*(T(\mathbf{x}_v, \mathbf{y}_{\mathcal{N}_u}))] \right\}
\end{aligned}$$

where $\mathcal{T}$ denotes any functions that map vertex and neighborhood features to a scalar, and the function $T(\cdot, \cdot)$ works as a variational representation of $t$. We further use an activation function $a : \mathbb{R} \to \mathrm{dom}_{f^*}$ to constrain the function value; that is $T(\cdot, \cdot) \to a(T(\cdot, \cdot))$. Therefore, we have

$$D_f(P_{\mathbf{v},\mathbf{n}}||P_{\mathbf{v}} \otimes P_{\mathbf{n}}) \;\geq\; \sup_{T \in \mathcal{T}} \left\{ \mathbb{E}_{P_{\mathbf{v},\mathbf{n}}}[a(T(\mathbf{x}_v, \mathbf{y}_{\mathcal{N}_v}))] - \mathbb{E}_{P_{\mathbf{v}}, P_{\mathbf{n}}}[f^*(a(T(\mathbf{x}_v, \mathbf{y}_{\mathcal{N}_u})))] \right\}$$

since the $a(T(\cdot, \cdot))$ is also in $\mathcal{T}$ and its value is in $\mathrm{dom}_{f^*}$, the optimal solution satisfies the equation. Suppose that the divergence function is $f(x) = x \log x$, the conjugate divergence function is $f^*(t) = \exp(t - 1)$ and the activation function is $a(x) = x$, we can obtain the $f$-representation of KL divergence; see Eq. (1). Note that the form of activation function is not unique, and we aim to find a proper one that helps to derivation and computation.

Here, we consider another form of divergence based on $f$-representation; that is, GAN-like divergence, where we have specific form of divergence function $f(x) = x \log x - (x + 1)\log(x + 1)$ and conjugate divergence function $f^*(t) = -\log(1 - \exp(t))$ [11]. We let the activation be $a(\cdot) = -\log(1 + \exp(\cdot))$. The GAN-like divergence is formulated as

$$\begin{aligned}
&D_{\mathrm{GAN}}\left(P_{\mathbf{v},\mathbf{n}}||P_{\mathbf{v}} \otimes P_{\mathbf{n}}\right) \\
\geq\; & \sup_{T \in \mathcal{T}} \left\{ \mathbb{E}_{P_{\mathbf{v},\mathbf{n}}}[a(T(\mathbf{x}_v, \mathbf{y}_{\mathcal{N}_v}))] - \mathbb{E}_{P_{\mathbf{v}}, P_{\mathbf{n}}}[f^*(a(T(\mathbf{x}_v, \mathbf{y}_{\mathcal{N}_u})))] \right\} \\
=\; & \sup_{T \in \mathcal{T}} \left\{ \mathbb{E}_{P_{\mathbf{v},\mathbf{n}}}[-\log(1 + \exp(-T(\mathbf{x}_v, y_{\mathcal{N}_v})))] + \mathbb{E}_{P_{\mathbf{v}}, P_{\mathbf{n}}} \log(1 - \exp(-\log(1 + e^{T(\mathbf{x}_v, \mathbf{y}_{\mathcal{N}_u})}))) \right\} \\
=\; & \sup_{T \in \mathcal{T}} \left\{ \mathbb{E}_{P_{\mathbf{v},\mathbf{n}}} \log \frac{1}{1 + e^{-T(\mathbf{x}_v, y_{\mathcal{N}_v})}} + \mathbb{E}_{P_{\mathbf{v}}, P_{\mathbf{n}}} \log(1 - \frac{1}{1 + e^{-T(\mathbf{x}_v, \mathbf{y}_{\mathcal{N}_u})}}) \right\} \\
=\; & \sup_{T \in \mathcal{T}} \left\{ \mathbb{E}_{P_{\mathbf{v},\mathbf{n}}} \left[ \log \sigma\left(T(\mathbf{x}_v, \mathbf{y}_{\mathcal{N}_v})\right) \right] + \mathbb{E}_{P_{\mathbf{v}}, P_{\mathbf{n}}} \left[ \log\left(1 - \sigma\left(T\left(\mathbf{x}_v, \mathbf{y}_{\mathcal{N}_u}\right)\right)\right) \right] \right\}
\end{aligned}$$

where $\sigma(\cdot)$ is the sigmoid function that maps a real value into the range of $(0, 1)$. Eventually, the GAN-like divergence converts the $f$-divergence to a binary cross entropy, which is similar to the objective function to train the discriminator in GAN [5].

To determine the form of the function $T(\cdot, \cdot)$, we parameterized $T(\cdot, \cdot)$ by trainable neural networks rather than design it manually. The parameterized function is denoted as $T_w(\cdot, \cdot)$, where $w$ generally denotes the parameterization. In this work, $T_w(\cdot, \cdot)$ is constructed with three trainable functions: 1) A vertex embedding function $\mathcal{E}_w(\cdot)$; 2) A neighborhood embedding function $\mathcal{P}_w(\cdot)$; and 3) a vertex-neighborhood affinity function $C_w(\cdot, \cdot)$; which are formulated as

$$\begin{aligned}
T_w(\mathbf{x}_v, \mathbf{y}_{\mathcal{N}_u}) &= \mathcal{S}_w(\mathcal{E}_w(\mathbf{x}_v), \mathcal{P}_w(\mathbf{y}_{\mathcal{N}_u})) \\
&= \mathcal{S}_w\left( \mathcal{E}_w(\mathbf{x}_v), \frac{1}{R} \sum_{r=0}^{R} \sum_{\nu \in \mathcal{N}_u} \left( (\widetilde{\mathbf{D}}^{-1/2} \widetilde{\mathbf{A}} \widetilde{\mathbf{D}}^{-1/2})^r \right)_{\nu, u} \mathbf{W}^{(r)} \mathcal{E}_w(\mathbf{x}_\nu) \right).
\end{aligned}$$

Table 1: The detailed information of graph datasets used in the experiments of graph classification

| Dataset | IMDB-B | IMDB-M | COLLAB | D&D | PROTEINS | ENZYMES |
|---|---|---|---|---|---|---|
| # Graphs | 1000 | 1500 | 5000 | 1178 | 1113 | 600 |
| # Classes | 2 | 3 | 3 | 2 | 2 | 6 |
| Max # Vertices | 139 | 89 | 492 | 5748 | 620 | 126 |
| Min # Vertices | 12 | 7 | 32 | 30 | 4 | 2 |
| Avg. # Vertices | 19.77 | 13.00 | 74.49 | 284.32 | 39.06 | 32.63 |
| # Train Graphs | 900 | 1350 | 4501 | 1061 | 1002 | 540 |
| # Test Graphs | 100 | 150 | 499 | 111 | 117 | 60 |
| Vertex Dimensions | 1 | 1 | 1 | 82 | 3 | 3 |
| Max Degrees | 66 | 60 | 370 | - | - | - |

where $\mathcal{E}_w(\cdot)$ is modeled by a Multi-layer perceptron (MLP), $\mathcal{P}_w(\cdot)$ is modeled by a $R$-hop graph convolution layer and $\mathcal{S}_w(\cdot, \cdot)$ is also modeled by an MLP. In $\mathcal{P}_w(\cdot)$, $\widetilde{\mathbf{A}} = \mathbf{A} + \mathbf{I}$ is the self-connected graph adjacency matrix and $\widetilde{\mathbf{D}}$ is the degree matrix of $\widetilde{\mathbf{A}}$; $\mathbf{W}^{(r)} \in \mathbb{R}^{d \times d}$ is the trainable weight matrix associated with the $r$th hop of neighborhood. The neighborhood embedding function $\mathcal{P}_w(\cdot)$ aggregates neighborhood information with in a geodesic distance threshold $R$. Note that $\mathcal{P}_w(\cdot)$ separately use neighborhood features $\mathbf{y}_{\mathcal{N}_u}$ in form of connectivity information and vertex features.

In this way, the GAN-like-divergence-based mutual information between graph vertices and neighborhoods can be represented with the parameterized GAN-like divergence, which is a variational divergence and works as a lower bound of of the theorical GAN-like-divergence-based mutual information; that is,

$$
\begin{aligned}
I_{\mathrm{GAN}}^{(\Omega)}(\mathbf{v}, \mathbf{n}) &= D_{\mathrm{GAN}}(P_{\mathbf{v},\mathbf{n}} \| P_{\mathbf{v}} \otimes P_{\mathbf{n}}) \geq \widehat{I}_{\mathrm{GAN}}^{(\Omega)}(\mathbf{v}, \mathbf{n}) \\
&= \max_w \left\{ \mathbb{E}_{P_{\mathbf{v},\mathbf{n}}} \left[ \log \sigma \left( T_w(\mathbf{x}_v, \mathbf{y}_{\mathcal{N}_v}) \right) \right] + \mathbb{E}_{P_{\mathbf{v}}, P_{\mathbf{n}}} \left[ \log \left( 1 - \sigma \left( T_w \left( \mathbf{x}_v, \mathbf{y}_{\mathcal{N}_u} \right) \right) \right) \right] \right\} \\
&= \max_w \frac{1}{|\Omega|} \sum_{v \in \Omega} \log \sigma(\mathcal{T}_w(\mathbf{x}_v, \mathbf{y}_{\mathcal{N}_v})) + \frac{1}{|\Omega|^2} \sum_{(v,u) \in \Omega} \log(1 - \sigma(\mathcal{T}_w(\mathbf{x}_v, \mathbf{y}_{\mathcal{N}_u})))
\end{aligned}
$$

Since we consider the dependency between vertices and neighborhoods within a specific vertex set, the possible outcomes of the joint distribution and the two marginal distributions are countable. We thus use the summation to aggregate all the possible cases. To maximize $\widehat{I}_{\mathrm{GAN}}^{(\Omega)}(\mathbf{v}, \mathbf{n})$ by training the internal function in $T_w(\cdot, \cdot)$, that is, $\mathcal{E}_w(\cdot)$, $\mathcal{P}_w(\cdot)$, and $\mathcal{S}_w(\cdot, \cdot)$, we can maximally approximate the mutual information between individual vertex and neighborhood for vertex selection in our VIPool. Note that the value of $\widehat{I}_{\mathrm{GAN}}^{(\Omega)}(\mathbf{v}, \mathbf{n})$ is not very close to the exact KL-divergence-based mutual information, but it has the consistency to $I^{(\Omega)}(\mathbf{v}, \mathbf{n})$ to reflect the pair of vertex-neighborhood with high or low mutual information, leading to effective vertex selection.

## Appendix B. Detailed Information of Experimental Graph Datasets

Here we show more details about the graph datasets used in our experiments of both graph classification and vertex classification. We first show the six datasets for graph classification in Table 1. We see that, we show the numbers of graphs, graph classes, vertices, numbers of graphs in training/test datasets and feature dimensions of all the six datasets. Note that, three social network datasets, IMDB-B, IMDB-M and COLLAB do not provide specific vertex features, where the vertex dimension is denoted as 1 and the maximum vertex degrees are shown in addition. In our experiments, we use one-hot vectors to encode the vertex degrees in these three datasets as their vertex features which explicitly contains the structure information.

We then show the details of three citation network datasets used in the experiments of vertex classification in Table 2. We see that, we present the numbers of vertices, edges, vertex classes and feature dimensions of the three datasets, as well as we show the separations of training/validation/test sets, where '# Train Vertices (full-sup.)' denotes the number of training vertices for full-supervised vertex classification and '# Train Vertices (semi-sup.)' denotes the number of training vertices for semi-supervised vertex classification.

Table 2: The detailed information of graph datasets used in the experiments of vertex classification

| Dataset | Cora | Citeseer | Pubmed |
|---|---|---|---|
| # Vertices | 2708 | 3327 | 19717 |
| # Edges | 5429 | 4732 | 44338 |
| # Classes | 7 | 6 | 3 |
| Vertex Dimension | 1433 | 3703 | 500 |
| # Train Vertices (full-sup.) | 1208 | 1827 | 18217 |
| # Train Vertices (semi-sup.) | 140 | 120 | 60 |
| # Valid. Vertices | 500 | 500 | 500 |
| # Test Vertices | 1000 | 1000 | 1000 |

Table 3: Vertex classification accuracies (%) of different methods, where 'full-sup.' and 'semi-sup.' denote the scenarios of full-supervised and semi-supervised vertex classification, respectively.

| Dataset<br># Vertices (Classes)<br>Supervision | Cora<br>2708 (7) | | Citeseer<br>3327 (6) | | Pubmed<br>19717 (3) | |
|---|---|---|---|---|---|---|
| | full-sup. | semi-sup. | full-sup. | semi-sup. | full-sup. | semi-sup. |
| DeepWalk [12] | $78.4 \pm 1.7$ | $67.2 \pm 2.0$ | $68.5 \pm 1.8$ | $43.2 \pm 1.6$ | $79.8 \pm 1.1$ | $65.3 \pm 1.1$ |
| ChebNet [3] | $86.4 \pm 0.5$ | $81.2 \pm 0.5$ | $78.9 \pm 0.4$ | $69.8 \pm 0.5$ | $88.7 \pm 0.3$ | $74.4 \pm 0.4$ |
| GCN [9] | $86.6 \pm 0.4$ | $81.5 \pm 0.5$ | $79.3 \pm 0.5$ | $70.3 \pm 0.5$ | $90.2 \pm 0.3$ | $79.0 \pm 0.3$ |
| GAT [13] | $87.8 \pm 0.7$ | $83.0 \pm 0.7$ | $80.2 \pm 0.6$ | $73.5 \pm 0.7$ | $90.6 \pm 0.4$ | $79.0 \pm 0.3$ |
| FastGCN [2] | $85.0 \pm 0.8$ | $80.8 \pm 1.0$ | $77.6 \pm 0.8$ | $69.4 \pm 0.8$ | $88.0 \pm 0.6$ | $78.5 \pm 0.7$ |
| ASGCN [8] | $87.4 \pm 0.3$ | - | $79.6 \pm 0.2$ | - | $90.6 \pm 0.3$ | - |
| Graph U-Net [4] | - | 84.4 | - | 73.2 | - | 79.6 |
| GXN | $\mathbf{88.9 \pm 0.4}$ | $\mathbf{85.1 \pm 0.6}$ | $\mathbf{80.9 \pm 0.4}$ | $\mathbf{74.8 \pm 0.4}$ | $\mathbf{91.8 \pm 0.3}$ | $\mathbf{80.2 \pm 0.3}$ |
| GXN (gPool) | $88.0 \pm 0.4$ | $84.4 \pm 0.6$ | $79.7 \pm 0.5$ | $74.4 \pm 0.6$ | $90.6 \pm 0.4$ | $79.8 \pm 0.4$ |
| GXN (SAGPool) | $87.8 \pm 0.6$ | $84.7 \pm 0.4$ | $80.0 \pm 0.5$ | $74.2 \pm 0.4$ | $90.9 \pm 0.3$ | $80.1 \pm 0.3$ |
| GXN (AttPool) | $88.4 \pm 0.3$ | $84.6 \pm 0.5$ | $80.6 \pm 0.4$ | $74.5 \pm 0.5$ | $91.3 \pm 0.3$ | $\mathbf{80.2 \pm 0.4}$ |

## Appendix C. More GXN Variants for Vertex Classification

Here we show more results of vertex classification of more variants of the proposed GXN associated with different pooling methods; that is, we test different pooling methods with the same GXN model framework, where the pooling methods include gPool [4], SAGPool [10] and AttPool [7]. The full-supervised and semi-supervised vertex classification accuracies of different algorithms on three citation networks are shown in Table 3. We see that, comprared to the previous pooling methods, the proposed GXN which uses VIPool could provide higher average classification accuracies for both full-supervised and semi-supervised vertex classification. Different GXN variants with different pooling methods tend to consistently outperform most state-of-the-art models for vertex classification, reflecting the effectiveness of the proposed GXN architecture.

## Appendix D. Effects of Neighborhood Radius for Vertex Classification

Here we show how the accuracy of vertex classification varies with different neighborhood radius $R$ in the vertex infomax pooling. We tune $R$ from 1 to 7, and we test our model for fully-supervised and semi-supervised vertex classifcation on both Cora and Citeseer datasets. The classification results are illustrated in Figure 1. We see that, the proposed model achieves the best vertex classification performance when $R$ equals 3 or 4. Thus we use $R = 3$ as the default hyper-parameter in our model.

## Appendix E. Use A Few Selected Vertices for Semi-supervised Vertex Classification Training

Here we consider active-sample-based semi-supervised classification, where we are allowed to select a few vertices and obtain their corresponding labels as supervision to train a classifier for vertex classification. In other words, we actively select training data in a semi-supervised classification task. Intuitively, since a graph structure is highly irregular, selecting a few informative vertices would potentially significantly improve the overall classification accuracy. Here we compare the proposed VIPool to random sampling. Note that for this task, we cannot compare with other graph pooling methods. The reason is that previous pooling pooling methods need a subsequent task to provide an explicit supervision; however, the vertex selection here should be blind to the final classification

Figure 1: Effects of $R$ for vertex classification

(a) Semi-supervised vertex classification on Cora.    (b) Semi-supervised vertex classification on Citeseer.

Figure 2: Comparison of semi-supervised vertex classification accuracies with a few selected and labeled data by using different vertex selection methods.

labels. The proposed VIPool is rooted in mutual information neural estimation and can be trained in either an unsupervised or supervised setting.

Specifically, given a graph, such as a citation network, Cora or Citeseer, we aim to show the classification accuracy as a function of the number of selected vertices. For example, there are 7 classes in Cora, we can select $7, 14, 21, 28$ and $35$ vertices ($1, 2, 3, 4$ and $5$ times of $7$) and use their ground-truth labels as supervision for semi-supervised vertex classification. As for Citeseer, there are 6 classes and we can select $6, 12, 18, 24$ and $30$ vertices. During evaluation, we test the performances on all the unselected vertices. We compare two method for vertex selection and classification: 1) the proposed **VIPool** method, where we use greedy algorithm to optimize $C(\Omega)$ for vertex selection; and 2) **Random Sampling**, where we randomly select each vertex with the same probability on the whole graph. We conduct semi-supervised vertex classification on the datasets of Cora and Citeseer. Figure 2 shows the the classification accuracies varying with the numbers of selected vertices for two vertex selection methods. We see that, when we select only a few vertices, such as fewer than 3 times of the number of vertex classes (i.e. $21$ for Cora and $18$ for Citeseer), the proposed VIPool method could select much more informative vertices than randomly sampling the same number of vertices, leading to over $10\%$ higher vertex classification accuracies. If we select more vertices by using the two vertex selection methods, the classification results corresponding to the two methods become closer to each other, indicating that a large number of selected vertices tend to potentially provide sufficient information to represent the rich patterns of the graphs and we could obtain more similar classification results than only selecting a few vertices.

## Appendix F. Illustration of Vertex Selection

To show the pooling effects of different pooling algorithms, we conduct a toy experiments to reconstruct three spatial mesh graphs with an encoder-decoder model. The encoder employs different pooling methods to squeeze the original graph into a few vertices (10 vertices) and the decoder attempt to reconstruct the original graphs based on the pooled vertex features and graph structures. To train the encoder-decoder model, we use an L2-norm loss to measure the distances between the vertex coordinates of reconstructed graphs and ground-truth graphs.

The three mesh graphs have vertex features as the 2D Euclidean coordinates and the specific vertex distributions are that 1) 88 vertices uniformly distribute in a circle region; 2) 503 vertices distribute in a hollow square region, where the vertices densely distribute around the center and sparsely distribute near the margins; 3) 310 vertices distribute in a circle, where the vertices densely distribute near the center and sparsely distribute around. The specific topologies are shown in the first row of Figure 3.

We compare the proposed VIPool operation with several baseline methods: random sampling, gPool [4], SAGPool [10] and AttPool [7]. The selected vertices are colored blue and illustrated in Figure 3. We see that, VIPool can abstract the original graphs more properly, where the preserved vertices distribute dispersely in both dense and sparse regions to cover the overall graphs. As for the baselines, we see that, 1) random sampling tends to select more vertices in dense regions, since each vertex is sampled with equal probability and the dense regions include more vertices and chances for vertex selection; 2) gPool and SAGpool calculate the importance weight for each vertices mainly based on vertex information itself without topological constraints, thus the selected vertices tends to distributed concentrated in local regions. 3) AttPool considers to model the local attentions and select more representative vertices, thus it can abstract graph structures to some extent, but the vertex distributions still slightly collapse the dense region.