[Reviews · NeurIPS 2020]

Review 1

Summary and Contributions: This paper proposes Graph Cross Networks (GXN) for modeling graph data. A GXN uses a vertex infomax pooling method to create multi-scale graphs, and then several feature crossing layers are used to enhance multi-scale information fusion. The authors conduct extensive experiments in both graph classification and vertex classification. The results show GXN outperforms many competitive baseline methods.

Strengths: 1. Important problem to the NeurIPS community. 2. Extensive experiment. 3. Good result.

Weaknesses: 1. Limited novelty. 2. Lack of comparison with existing methods.

Correctness: The claims are correct.

Clarity: Overall, the paper is well written and easy to follow.

Relation to Prior Work: The paper does not have a section of related work, and there is only a single paragraph in introduction that talks about the relation to existing graph pooling methods, which is not very clear to me.

Reproducibility: Yes

Additional Feedback: This paper proposes GXN for modeling graph data. Overall, the proposed approach is quite intuitive, and extensive experiment on graph classification and vertex classification proves the effectiveness of the approach. I have the following concerns regarding the paper: 1. The relation to existing graph pooling techniques is not clear. One key component of GXN is vertex informax pooling, which is used to create multi-scale graphs and is thus related to existing graph pooling techniques. However, this paper does not contain a section of related work, and the authors only talk about the relation to existing pooling methods in introduction. As there are so many graph pooling papers recently, I wonder what is the advantage or contribution of vertex informax pooling over existing pooling methods. It would be helpful if the authors could further explain that during rebuttal. 2. Technically, the novelty of the paper is limited. In GXN, two key ideas are vertex informax pooling and feature-crossing layers. For vertex infomax pooling, GXN proposes to use mutual information estimation and maximization techniques for graph pooling. However, these techniques have been extensively used in graph machine learning, such as Deep Graph Infomax. Given these works, although the idea of using infomax for graph pooling seems interesting, technically the contribution is quite limited. For the feature-crossing layer, it looks more like a straightforward extension of Graph U-net, where graphs of multiple scales can interact with each other. From this sense, the contribution of this part is quite marginal. Overall, despite some interesting ideas in the paper, I feel like the contribution is not sufficient. 3. I also have some questions regarding the experiment. (1) For vertex classification, as the three datasets Cora, Citeseer, Pubmed are quite small, people usually run each model with different seeds (e.g., 50 or 100) to report the mean and standard deviation. In experiment, how many runs did the author use to compute the statistics of GXN? (2) For vertex classification, only the results on the standard data splits are reported. To better convince readers, it would be helpful to evaluate GXN on more random splits. ------------------------------ Thanks the authors for the detailed explanation, and all the additional results on vertex classification! I still have a few concerns on the VIPool method, and hope the authors could address them in the revised paper. (1) In VIPool, P_v, P_n, P_{v,n} are all discrete distributions (although the feature vector can be continuous, as there are |V| nodes, the sample from P_v can only have at most |V| values, so it is a discrete distribution). For such discrete distributions, the mutual information can be directly computed according to the definition in O(|V|^2) time without using those neural estimators. Although the neural estimator may give smoother estimation, they also have much larger computational cost, so I don't see why it is necessary to use these neural estimators for this task. (2) The authors claimed that the cost of VIPool is O(|V|). From my understanding, VIPool greedily selects |\Omega| nodes, where |\Omega| is proportional to |V|. To select each node, VIPool needs to draw negative samples from the node set, so the cost is O(|V|). Therefore, the total cost is at least O(|V|^2) rather than O(|V|). (3) To speed up pooling, the authors mentioned that negative sampling was used, but the number of negative samples and the distribution of negative samples are not clearly mentioned, which are important to replicate the results of the paper. Despite the above concerns, I feel like the authors have addressed most of my concerns, and the idea of using mutual information for pooling is quite new. Therefore, I raised my score to weak accept.


Review 2

Summary and Contributions: A new graph pooling operator based on the mutual information between vertices and their neighborhood is proposed and used in a new GNN architecture supporting information flow between different levels of abstraction.

Strengths: 1) Well-engineered approach with some original contributions. 2) Clearly written, relation to other work is made explicit. 3) Solid experimental evaluation showing the merits of the approach comparing to state-of-the-art approaches.

Weaknesses: 1) Pooling has been studied extensively and this work clearly is incremental building on known techniques. 2) I would like to see a more detailed discussion of the choice of the parameter K.

Correctness: Yes, to the best of my knowledge.

Clarity: The paper is well-written and supported by figures. The paper is quite condensed and on a good technical level, but the authors managed to keep it easily readable and comprehensible.

Relation to Prior Work: The relation to other work is clearly stated and the differences are discussed in detail. The relevant literature is cited as far as I can tell.

Reproducibility: Yes

Additional Feedback: In the experimental evaluation it is stated that "To improve the efficiency of solving problem (1), we modify C(Ω) by preserving only the first term." -- I do not understand this; probably the wrong equation is referenced here. This approach should be justified. ==== Update after the rebuttal ==== Thank you for providing additional experimental results. I have left my score unchanged.


Review 3

Summary and Contributions: In this paper, the authors propose a vertex pooling method based on neural information selection and then design a new architecture to compress the graph. The experimental results show moderate improvement over the existing approaches.

Strengths: S1: The definition of selecting vertices with mutual information is novel and interesting S2: The experimental results show reasonable improvement.

Weaknesses: W1: the insight of the proposed pooling method compared with other pooling is lacking W2: the performance improvement of the proposed pooling compared with exiting methods are not significant. Especially, in a few datasets, structurePool method seems to be better than GXN with other pooling. What if you combine StructPool with GXN?

Correctness: Yes.

Clarity: Reasonable

Relation to Prior Work: I would like the authors provide a more detailed review on the state-of-the-art pooling compared with the new approach.

Reproducibility: Yes

Additional Feedback:


Review 4

Summary and Contributions: This paper proposes graph cross network to achieve comprehensive feature learning from multiple scales of a graph. The two key components of graph cross network include a novel vertex infomax pooling, which creates multiscale graphs in a trainable manner and a novel feature crossing layer, enabling feature interchange across scales. The proposed VIPool selects the most informative subset of vertices based on the neural estimation of mutual information between vertex features and neighborhood features. The proposed work has been used for graph as well as node classification tasks where it obtained state-of-the-art results.

Strengths: The paper proposed two novel techniques (1) vertex infomax pooling which is basically a hierarchical graph pooling considering the neighborhood of nodes and (2) graph cross network for multiscale feature learning, which I have found novel compared to the existing literature. The paper also experimentally compared existing works with the proposed one, which made it quite strong.

Weaknesses: Although the experiments demonstrated in paper are quite convincing, some further experiments would have been interesting for the community: (1) Effect of neighborhood hops R for vertex classification accuracies (2) The comparison of training times with existing similar techniques These experiments can either be added to the supplementary or in a future extended version.

Correctness: I think the claims and method is absolutely correct. The experimental results are convincing.

Clarity: The paper is quite well written and organized. It was easy to follow.

Relation to Prior Work: I find one existing existing paper is missing in the reference: [1] Rex Ying, Jiaxuan You, Christopher Morris, Xiang Ren, William L. Hamilton, Jure Leskovec, Hierarchical Graph Representation Learning with Differentiable Pooling, NeurIPS, 2018. Others are properly mentioned as far as I understand.

Reproducibility: Yes

Additional Feedback: (1) The overall loss function of the method is written as L = L_{task} + \alpha L_{pool}, I am curious about the behavior of \alpha and also the stability of the training procedure. It would be interesting to see some kind of plot about the behavior of the mutual information with the number of training iterations. (2) Is it possible to use vertex infomax pooling together with other kind of architectures if multiscale graph neural networks, such as Graph U-net. I think an experiment demonstrating that would be interesting to the community. (1) The graph cross network has feature crossing layers which I suppose to be computationally very expensive, and hence it should be difficult to train. It would be interesting to hear some information on the training procedure of the model. (2) I am also curious about the effect of number of layers and hierarchical levels used to train the model. Apart from these comments, I am quite happy with the paper. After rebuttal: ========== I have read through the rebuttal submitted by the authors and I am quite satisfied with that, hence I am happy to accept the paper.

[Author Response · NeurIPS 2020]

We appreciate insightful comments from all reviewers to our paper 'Graph cross networks with vertex infomax pooling'.

First, we address three common concerns about the proposed vertex infomax pooling (VIPool) and GXN.

• **VIPool vs. Other graph pooling. 1.** VIPool is a novel method for vertex selection, which is also critical to network

science, graph theory and graph signal processing. Recent graph pooling methods mainly have two approaches: the

vertex-grouping-based approach (DiffPool [47] and StrucPool [48]), which groups vertices to some clusters; and the

vertex-selection-based approach (gPool [20], SAGPool [29], AttPool [25]), which selects representative vertices and

then coarsens the graph based on the selected vertices. **2.** VIPool provides an *explicit optimization* (Eq.1) for vertex

selection, which can be trained via self-supervision. Most vertex-selection methods, including gPool, SAGPool and

AttPool, purely rely on a subsequent task to select vertices, lacking generalization and interpretation. For example, only

VIPool can be used in active sampling for semi-supervised learning; see Appendix E. **3.** VIPool *resolves the clustering*

*issue* in many vertex-selection-based approaches, including gPool and SAGPool; see Appendix F. The clustering issue

is: most selected vertices come from a small subgraph. In VIPool, the vertex-selection criterion explicitly punishes

those selected vertices that share similar neighborhoods. **4.** Compared to recent vertex-grouping-based approaches,

VIPool has a *lower computational cost* than StrucPool ($O(N)$ vs. $O(N^3)$); DiffPool requires a subsequent task to

supervise vertex clustering, while VIPool has an explicit optimization to select vertices.

• **VIPool vs. Deep graph infomax (DGI).** Both leverage mutual information neural estimation (MINE) [2]. Two

major differences are: **1. Aim.** VIPool aims to obtain an optimization for vertex selection whose objective function is

obtained through MINE; while DGI aims to learn a *graph embedding*, which is a trainable mapping updated through

MINE. **2. Formulation.** Since VIPool selects vertices in a given graph, VIPool trains on *a single graph* and its training

samples are positive/negative *pairs of vertices and neighborhoods* in the same graph; since DGI maps each graph to an

embedding, DGI trains on *multiple graphs* and its training samples are positive/negative *pairs of vertices and graphs*.

• **GXN vs. Graph U-Nets.** Two major differences are: **1. Intermediate fusion vs. late fusion.** GXN fuses features at

multiple scales in each network layer while graph U-net fuses features at the end of each scale. **2. Deep vs. shallow**

**learning in each scale.** GXN extracts features multiple times in each scale while graph U-net extracts single-scale

features only once in each scale and then uses a skip-connection to fuse features across scales.

Next, we address the specific questions from each reviewer. **Please zoom in to see the figures precisely.**

• **Reviewer 1:** Q1: Compare VIPool to previous methods. A1: see VIPool vs. Other graph pooling.

Q2: VIPool is similar to DGI. GXN is a straightforward extension of graph U-net. A2: We researchers all build our works

on the shoulders of giants and we pursue simple, yet nontrivial designs. VIPool and GXN make distinct contributions

to self-supervised trainable vertex selection and multiscale architecture design, respectively; see VIPool vs. DGI and

GXN vs. Graph U-Nets. Compared to previous works, both designs are nontrivial, effective and intuitive.

Q3: How many runs for vertex classification with different model initialization? Try other dataset splits. We run 100

times with different initializations. We test 5 random splits and compare GXN to GCN and GAT. The results of the

semi-supervised vertex classification on Cora are GCN/GAT/GXN: $78.4 \pm 0.7/79.7 \pm 1.3/\mathbf{81.4 \pm 0.8}$

• **Reviewer 2:** Q1: VIPool builds on known techniques. A1. see VIPool vs. Other graph pooling and VIPool vs. DGI.

Q2: Effects of the numbers of selected vertices $|\Omega|$. A2. Fig. 1 (a) shows the appropriate and effective $|\Omega|$ is important.

Q3: Why do we modify $C(\Omega)$ to solve (1)? A3. The modification makes the method faster without sacrificing too much

performance. On IMDB-B: before modification: $77.7 \pm 0.5\%$ accuracy, $3.7 \times 10^{-2}$ $second$ test time cost per graph;

after modification: $77.3 \pm 0.8\%$ accuracy, $4.3 \times 10^{-5}$ $second$ test time cost per graph, which is much faster.

• **Reviewer 3:** Q1: Not enough comparisions of VIPool to other methods. A1. see VIPool vs. Other graph pooling.

Q2: Combine StructPool to GXN. A2. Table 1 shows VIPool outperforms StructPool on graph and vertex classification.

| | IMDB-B | IMDB-M | COLLAB | DD | PROTEINS | ENZYMES | Cora | Citeseer | Pubmed |
|---|---|---|---|---|---|---|---|---|---|
| GXN-StructPool | 76.40 | 54.02 | 79.35 | 83.77 | 80.03 | **60.17** | 84.4 | 74.2 | 79.8 |
| GXN-VIPool | **77.30** | **54.57** | **80.62** | **84.26** | **80.38** | 59.59 | **85.1** | **74.8** | **80.2** |

Table 1: Based on the same GXN architecture, we compare StructPool and VIPool on graph and vertex classification.

• **Reviewer 4:** Q1: Effects of neighborhood radius $R$ for vertex classification. A1. Fig. 1 (b) shows that various $R$s are

stale and lead to minor effects for vertex classification. We choose $R = 3$ in our model.

Q2: Compare the training time, show the training process. A2. Fig. 1 (c) shows both task and pooling losses converge

stably; the overall loss descends with $\alpha$; and GXN converges faster than StructPool and graph U-net.

Q3: Effects of $\alpha$ and mutual information in training. A3. Fig. 1 (d) shows the training loss converges stably with

various $\alpha$. We initialize $\alpha = 2$ to balance task objective minimization and mutual information maximization.

Q4: VIPool on other architectures. A4. On IMDB-B: Encoder-decoder+VIPool: $74.0 \pm 1.0\%$; Readout+VIPool:

$76.3 \pm 0.9\%$; Graph U-net+VIPool: $76.7 \pm 0.5\%$; GXN+VIPool: $\mathbf{77.3 \pm 0.8}\%$. GXN outperforms the others.

Q5: Show how difficult to train feature-crossing and effects of number of layers. A5. Fig. 1 (e) shows although fewer

feature-crossing layers converge faster, training feature-crossing layers is not hard.

Figure 1: (a) Effect of $|\Omega|$; (b) Effect of $R$; (c) Training loss; (d) Effect of $\alpha$; (e) Effect of feature crossing and hidden layers.

[Meta-Review · NeurIPS 2020]

The paper make a novel contribution by introducing graph cross networks, and demonstrate it usefulness in practical example. While initial concern related to the clarity of the paper, the reviewers found that the authors have done a good job in summarizing their work and addressed most of their concerns in the rebuttal. The two key components of GXN are a novel vertex infomax pooling, which creates multiscale graphs in a trainable manner and a novel feature crossing layer, enabling feature interchange across scales. This work has been compared their work with prior methods and surpassed all of them, which meets the bar for a NeurIPS presentation. While it does not impact the decision, during the discussion, the following points were left unanswered, and it would be great if the authors could take the following points in their reviews: (1) In VIPool, P_v, P_n, P_{v,n} are all discrete distributions (although the feature vector can be continuous, as there are |V| nodes, the sample from P_v can only have at most |V| values, so it is a discrete distribution). For such discrete distributions, the mutual information can be directly computed according to the definition in O(|V|^2) time without using those neural estimators. Although the neural estimator may give smoother estimation, they also have much larger computational cost, so it seems important to better explain why these neural estimators intuitively much better for this task (2) The authors claimed that the cost of VIPool is O(|V|) in their response. Some people understand that it also imply that VIPool greedily selects |\Omega| nodes, where |\Omega| is proportional to |V|. To select each node, VIPool needs to draw negative samples from the node set, so the cost is O(|V|). Therefore, the total theoretical cost is at least O(|V|^2) rather than O(|V|). (3) To speed up pooling, the authors mentioned that negative sampling was used, but the number of negative samples and the distribution of negative samples were not clearly mentioned, which are important to replicate the results of the paper.